# Healthcare Expenditures among the Elderly in China: The Role of Catastrophic Medical Insurance

**DOI:** 10.3390/ijerph192114313

**Published:** 2022-11-02

**Authors:** Hongmei Cao, Xinpeng Xu, Hua You, Jinghong Gu, Hongyan Hu, Shan Jiang

**Affiliations:** 1Affiliated Drum Tower Hospital, Medical School of Nanjing University, Nanjing 210008, China; 2Center for Health Policy and Management Studies, Nanjing University, Nanjing 210093, China; 3Institute of Healthy Jiangsu Development, Nanjing Medical University, Nanjing 211166, China; 4School of Public Health, Nanjing Medical University, Nanjing 211166, China; 5College of Art and Sciences, University of Washington, 1400 NE Campus Parkway, Seattle, WA 98105, USA

**Keywords:** healthcare expenditure, catastrophic medical insurance, the elderly, China

## Abstract

China has been piloting the catastrophic medical insurance (CMI) program since 2012 and rolled it out nationally in 2016 to reduce the incidence of catastrophic health expenditure among Chinese residents. Few studies have been conducted to determine its effect on healthcare expenditures, particularly among the elderly. The purpose of this study is to examine the effect of CMI on healthcare expenditures among China’s elderly population. The data for this study were derived from 4 waves of the Chinese Longitudinal Healthy Longevity Survey, which included 344 and 1199 individuals in the treatment and control groups, respectively. To examine the effect of CMI on healthcare expenditures among the elderly, we used difference-in-differences and fixed-effects models. Additionally, a heterogeneity analysis was used to examine the differences in the impact of CMI on different groups. Finally, we confirmed the robustness of the results using robustness and placebo tests. CMI increased total health and out-of-pocket expenditures significantly, as well as inpatient and corresponding out-of-pocket expenditures. The reassults of the heterogeneity analysis indicated that CMI had a greater impact on elderly residents of rural areas. Economic burden protection has been enhanced for low-income groups and patients with serious diseases over the last two years. Our research indicated that CMI can promote the use of inpatient medical services for the elderly to a certain extent. Targeted measures such as expanding the CMI compensation list, establishing a more precise compensation scheme, and specific diseases associated with high healthcare expenditures can be considered in the practice of CMI implementation.

## 1. Introduction

To enhance residents’ financial protection and access to healthcare, China has developed a basic medical insurance system over the last two decades, including urban employee basic medical insurance (UEBMI) for urban employees, urban resident basic medical insurance (URBMI) for urban unemployed residents, and the New Cooperative Medical Scheme (NCMS) for rural residents [1,2], and achieved nearly full coverage in a relatively short period of time [3]. Basic medical insurance can increase access to and utilization of medical services [4]. However, there is a disparity in compensation list and reimbursement rates between different medical insurance schemes, with URBMI and NCMS participants paying a higher co-payment rate than UEBMI participants [5]. Due to insufficient financing and low reimbursement rates, basic health insurance plays a limited role in mitigating catastrophic health expenditures among residents, particularly those enrolled in URBMI and NCMS [6,7,8].

China proposed launching a pilot program for catastrophic medical insurance (CMI) in 2012 to further alleviate residents’ exposure to catastrophic health expenditures [9]. There were 134 city pilots in 2013, and other cities were completely covered by 2016. CMI is a mandatory insurance for NCMS and URBMI participants. Individuals insured under URBMI and NCMS would be automatically enrolled in the CMI scheme without additional payment [10]. Following consultation with a physician, a participant is compensated by their basic medical insurance. If an individual continues to have significant out-of-pocket (OOP) expenses after reimbursement for basic medical insurance, the portion of these out-of-pocket expenses that are eligible for CMI reimbursement (referred to as compliance medical costs) would be reimbursed in addition [11]. Local governments determine the deductibles (thresholds) for CMI based on the per capita disposable income of urban residents and the per capita net income of rural residents in the preceding year. Similarly, local governments set the reimbursement rates and ceilings. Reimbursement rates increase in lockstep with the increase in out-of-pocket medical costs associated with compliance. CMI covers only inpatients in almost all regions, which means that it pays only those inpatients whose cumulative compliance expenses over the previous year exceeded the CMI threshold after reimbursement by basic medical insurance. According to China Banking and Insurance Regulatory Commission (CBIRC) data, CMI reached 100% coverage of the target population in 2016, covering 1.07 billion urban and rural residents, and its reimbursement ratio based on basic medical insurance increased by 14% in 2020 [12].

Intuitively, the introduction of CMI has enhanced the financial protection of urban and rural residents, but the specific impact of CMI on residents’ health expenditures needs to be investigated further. This study aims to discuss the following issues: What effect would CMI have on elderly health expenditures and out-of-pocket expenses? How does the impact of CMI change for different groups (urban and rural populations, different income groups, and whether they have had a serious illness within the prior two years)? We analyzed the potential impact on CMI based on Anderson’s health service utilization model and conducted empirical analyses on the aforementioned issues using the representative dataset of the Chinese Longitudinal Healthy Longevity Survey (CLHLS) at the national level, so as to enrich the research results of the implementation effect of CMI and provide empirical evidence for further improving and optimizing CMI system design in practice.

## 2. Literature Review and Research Hypothesis

### 2.1. Literature Review

The majority of current research on CMI falls into the following categories. Certain studies compared CMI policy documents published in various regions, identifying shortcomings in the design and implementation of welfare packages and making recommendations for improvement [13,14]. By simulating various CMI compensation schemes, some studies compared the effects of CMI under various catastrophic health expenditure measurement methods and compensation schemes [15,16,17,18,19]. By examining the effectiveness of CMI implementation using datasets from health insurance information systems and survey data at some regional or national levels, some studies discovered that increased CMI coverage significantly effectively alleviated poverty caused by illness [20], and reduced vulnerability to poverty and broke the vicious cycle of poverty and disease through increased income [21].

However, some studies found that, while CMI did alleviate the economic burden of patients with high medical costs, the effect was not sustained over time [22], and wealthy families benefited more from CMI [23]. Although CMI increased reimbursement rates following implementation, its impact on reducing the incidence of catastrophic health expenditure was limited [24,25]. While CMI decreased the incidence of catastrophic health expenditure, it increased the intensity of catastrophic health expenditure [26]. A study of rural residents discovered that CMI was insufficient in relieving rural residents’ medical burdens [27]. Due to the high out-of-pocket costs, individuals with better economic circumstances could receive CMI compensation, resulting in unfair benefits for CMI [28]. CMI increased patients’ inpatient and outpatient medical expenses, but also their out-of-pocket expenses [29]. Zhao discovered that implementing CMI had no effect on rural residents’ healthcare expenditures [30]. Along with the protective effect of economic burdens, some studies were conducted on the effect of CMI on health. CMI has been shown in studies to improve self-reported health in middle-aged and elderly residents [31], as well as rural residents [32], and to decrease the mortality rate of the elderly [33].

As can be seen from the existing research, the evaluation of the implementation effect of CMI has not yet resulted in a consistent conclusion. Few studies examined the impact of CMI on healthcare expenditures and out-of-pocket expenses using national data. The primary objective of this study is to determine the effect of CMI on health expenditures and the heterogeneity of the impact on different groups. The elderly were chosen as the research subjects primarily because their health status is poorer, their prevalence is higher, and their demand for healthcare is greater [34]. China has the world’s largest elderly population and the world’s fastest growing population [35]. According to China’s seventh national census, the number of people aged 60 or over in the country reached 264 million by 2020, accounting for 18.4 percent of the total population [36]. Healthcare costs will continue to rise as population aging accelerates, imposing a significant burden on society. Additionally, because older adults spend more on healthcare than other adults, their demand for healthcare services is expected to be more elastic [37]. That is, they are more sensitive to changes in the cost of healthcare, and the impact on this group may be greater when evaluating the effects of health insurance [38].

Compared to previous studies, the following were the primary marginal contributions of this study: (1) Using panel data to analyze the causal effect of CMI implementation on health expenditures for the elderly as the research object. In contrast to a large number of previous studies that used the claims data of the health insurance information system from the health insurance bureau to conduct comparative analyses before and after the implementation of CMI, this study utilized the gradual implementation of CMI as a quasi-natural experiment. The nationally representative CLHLS dataset was utilized to study the influence of CMI on health expenditures among the elderly, who are particularly sensitive to changes in healthcare service prices. (2) Heterogeneity analysis was utilized to analyze the impact of CMI implementation on different groups, and the CMI implementation effect was examined from a broader perspective. Previous studies focused mostly on the effects of CMI on residents of a particular area or rural residents, but this study included other CMI-affected populations. The heterogeneity analysis of subsamples accurately showed the impact of CMI on different groups and provided the basis for further policy design adjustments. (3) A set of robustness tests was conducted to verify that the study results were more accurate and reliable. Considering the possibility of estimation bias, we tested the robustness of our results using a variety of techniques, such as excluding confounding sample regions, employing a fixed-effects model based on propensity score matching, event study analysis, and placebo testing, which serve as a benchmark for ensuring the accuracy of effects estimation in similar policies.

### 2.2. Research Hypothesis

Utilization of healthcare services is a prerequisite for healthcare expenditures. According to the Anderson’s health service utilization model, the main determinants of an individual’s healthcare utilization are predisposing characteristics, enabling resources, and the need. Predisposing characteristics represent the individual tendency to utilize health services, including individual demographic characteristics, social structure, as well as health beliefs. The enabling resources refer to the resource endowment of individuals to utilize health services, including individual and family resources, as well as community resources. Need primarily refers to the direct influencing aspects of an individual’s utilization of health services, that is, health status, including the individual’s perceived need and evaluated need for utilization of health services by medical staff [39].

#### 2.2.1. Predisposing Characteristics

The healthcare service utilization model assumes that people with certain characteristics prior to the onset of disease are more likely to use health services. These characteristics include demographic characteristics such as age and gender, as well as characteristics that reflect an individual’s social status, such as education level, which is closely related to the individual’s lifestyle and healthcare-related behavior, which in turn affects the individual’s use of health services. Although it is not a direct cause of health service utilization, health beliefs are also regarded to be a predetermined element that influences an individual’s utilization of health services. People with more favorable attitudes and opinions toward healthcare are more likely to utilize health services. Additionally, previous illness is regarded as one of the risk factors. People having a history of health issues are more likely to require future healthcare services.

#### 2.2.2. Enabling Resources

To access healthcare services, individuals need to have certain favorable conditions, which can be determined by family resources. For instance, household income and health insurance coverage could enable individuals to utilize healthcare services. In addition, community-level resources, such as urban and rural residence, community health resources, and the price of health services, would influence the utilization of individual health services.

#### 2.2.3. Need

Health status is the most fundamental reason why persons seek treatment. This encompasses the individual’s own perception and experience of illness status or self-reported overall health level, as well as the individual’s disease diagnosis and severity as determined by disease evaluation measures.

CMI aims to further improve residents’ level of medical security, to alleviate their financial burden, to protect their health rights and interests, and to avoid the phenomenon of illness-induced poverty or relapse into poverty by providing secondary compensation for the high medical expenses incurred after basic medical insurance reimbursement. Obviously, CMI is an enabling resource that affects the utilization of individual health services. The conceptual framework for this study is depicted in Figure 1. Through secondary compensation, CMI implementation can further reduce the cost of medical services for individuals. Reduced medical service prices frequently result in increased demand for health services [40,41]. The health service demand is defined as the quantity of medical services that people are able and willing to consume during a specified time period at a specified price level, including demand derived from need and demand without need. The former is referred to as efficient demand, whereas the latter is frequently caused by individual excess demand and induced demand from health providers. Both types of demand for health services will result in an increase in healthcare expenditures, and as a result, we anticipate that CMI would result in an increase in healthcare expenditures for the elderly.

Health insurance’s degree of protection against the insured’s economic risk is primarily determined by its compensation list and reimbursement ratio. If CMI could offset the increase in health expenditure, CMI would have no effect on or even reduce OOP. However, if CMI was unable to offset the increase in health expenditure caused by increased healthcare utilization, CMI would increase patients’ out-of-pocket expenditures. Individuals will have the right to receive treatment of a higher quality if financial protection is strengthened. The implementation of CMI would considerably enhance the willingness of hypertension and diabetic patients to seek treatment in urban tertiary health facilities, according to a study from a county in China (Tongxiang County) [29]. The introduction of CMI relaxed their restrictions on medical treatment at high-quality health facilities, which increased their out-of-pocket payments. Considering previous research, Hypothesis 1 is proposed:

**Hypothesis 1 (H1).** *CMI will increase total healthcare expenditures and inpatient expenditures. Implementation of CMI would increase out-of-pocket medical expenses of individuals*.

As stated previously, in addition to examining the overall impact of CMI on the healthcare expenditures of the elderly, we desire to understand the differences in the implementation effects of CMI on different population groups. Here, we chose three factors for subgroup heterogeneity analysis that are intimately related to the design of CMI policies. The first variable of interest is the residence. In China, a person’s participation in what kind of health insurance depends on their residence, employment, and household registration. Residents in rural areas could voluntarily choose to participate in the NCMS, whereas unemployed residents in urban areas could voluntarily choose to participate in the URBMI. URBMI and NCMS insureds are covered by CMI. Due to changes in the design of the basic medical insurance system, the ratio of reimbursement for rural residents’ basic medical insurance is smaller, and the list of compensation is more limited than those in URBMI. Therefore, rural residents are closer to the reimbursement threshold of CMI due to their greater out-of-pocket costs after reimbursement from basic medical insurance. We anticipate that this population would be more susceptible to CMI than urban residents. We therefore propose Hypothesis 2:

**Hypothesis 2 (H2).** *CMI will have a significantly greater effect on rural residents than on those who live in cities or towns*.

The second variable we focused on is income. It is widely known that income has a significant role in determining individual demand for health services. The incentive effect of health insurance on medical service consumption varies according to income group. Due to the fact that low-income groups have a larger price elasticity of demand for medical services than high-income groups [42], healthcare expenditures in this group are more likely to be affected by price changes in medical services. It is anticipated that CMI would have a bigger impact on low-income groups due to the introduction of CMI, which enhanced financial protection and relaxed budgetary limits while seeking medical treatment. Consequently, Hypothesis 3 is proposed:

**Hypothesis 3 (H3).** *The effect of CMI on low-income groups will be stronger than that on high-income groups*.

The third variable of interest is whether an individual has suffered a serious illness during the preceding two years. The original intention of CMI was to relieve the high medical economic burden of residents by secondary compensation, hence decreasing the likelihood of catastrophic health expenditures. As a factor that determines whether a person is more likely to use health services, those with serious illnesses in the past two years are also potential poverty targets owing to illness. CMI is theoretically implemented to mitigate the economic risk of individuals associated with serious diseases. As a result, it is self-evident that CMI’s effect on the elderly with serious diseases in the preceding two years would be greater. As a result, we propose Hypothesis 4:

**Hypothesis 4 (H4).** *CMI will have a significantly greater effect on patients with serious illness than on those who had not had a serious illness in the preceding two years*.

## 3. Data, Measures, and Identification Strategy

### 3.1. Data Source

This study used data from the Chinese Longitudinal Healthy Longevity Survey (CLHLS), a large-scale longitudinal survey conducted jointly by the China Center for Economic Research of Peking University, Center for Healthy Aging, and Family Studies of Peking University and Duke University [43]. The survey’s objective is to examine the elderly’s quality of life, healthy aging, and the factors that influence their longevity. CLHLS began in 1998 with a baseline survey of people aged 65 and over in 801 counties spread across 23 provinces (autonomous regions or municipalities). The survey was then repeated in 2000, 2002, 2005, 2008, 2012, 2014, and 2018. The CLHLS contains information about the elderly’s social and demographic characteristics, family information, health status, medical insurance, and healthcare expenditures, which lays a foundation for the study.

We hoped to retain as large a sample as possible, which will enable us to not only accomplish the research objective, but also to test the research method’s premise to ensure the reliability of the research results. As a result, we retained the elderly from 2008, 2012, 2014, and 2018. We examined the parallel trend assumption using data from the preceding four periods. However, because the CMI implemented full coverage after 2016, and the integration of URBMI and NCMS has been implemented nationally since 2016, we excluded data from 2018 from the formal analysis of the impact of the CMI on healthcare expenditures for the elderly and analyzed only the data from the previous three phases. Finally, 1543 elderly people were interviewed in 4 phases between 2008 and 2018, with 344 and 1199 individuals in the treatment and control groups, respectively (Figure A1).

### 3.2. Measures

#### 3.2.1. Healthcare Expenditures

Consider that CMI provides additional compensation to patients based on reimbursement from URBMI and NCMS, and that CMI’s primary target group is hospitalized patients. Thus, total healthcare expenditures over the previous year and out-of-pocket expenditures, as well as inpatient healthcare expenditures and out-of-pocket expenditures, were used for measuring healthcare expenditures.

#### 3.2.2. Catastrophic Medical Insurance (CMI)

In this study, the key independent variable is a dummy variable indicating whether CMI was implemented. When the value is 1, it indicates that CMI has been implemented. CMI was not implemented when it was equal to 0. According to a previous study [32], the provinces of Liaoning, Jilin, Zhejiang, Hubei, and Fujian, as well as the municipality of Chongqing, were considered the treatment group (where CMI was implemented in 2013), while the remaining provinces (and municipalities) were considered the control group (where CMI had not been implemented in 2013).

#### 3.2.3. Covariates

Considering factors affecting health service demand and health expenditures, existing research analyzes factors affecting healthcare utilization primarily through the Anderson healthcare utilization conceptual framework [44,45]. As mentioned above, this framework was used to identify covariates associated with health expenditures in this study. The framework identifies three types of indicators as influencing healthcare utilization: predisposing, enabling, and need variables [46]. As a result, these variables were included as covariates in this study. Age, gender, marital status, number of surviving children, living arrangement, and whether the elderly had a serious illness in the previous two years were all predisposing variables. We did not include the health belief as a predisposing predictor for two reasons. First, since we cannot collect health belief-related variables from CLHLS, and second, as noted by Anderson and Newman, there is a correlation between demographic and social structure features and health beliefs [39], and we control for these characteristics in the model. Along with the key independent variables, the enabling variable included the type of medical insurance, residence, and household income per capita. Finally, to account for the effect of health on the health service demand, the study included three kinds of indicators for the need variable: self-rated health, the prevalence of chronic diseases, and specific diagnosis among the elderly.

### 3.3. Identification Strategies

#### 3.3.1. Difference-in-Differences Regression Model

The difference-in-differences (DID) model was used as a baseline for analyzing the effect of CMI on elderly healthcare expenditures. The model was set as follows:(1)$$y_{ipt}=\alpha +\beta \cdot treat_{ipt}\times Post_{it}+\gamma \cdot treat_{ipt}+\delta \cdot Post_{it}+Z_{ipt}^{\prime }\eta +\sigma_{p}+\lambda_{t}+\varepsilon_{ipt}$$
where $y_{ipt}$ represents the healthcare expenditures of individual i in province p during period t. $treat_{ipt}$ represents the dummy variable for whether CMI was implemented in province p, and the value of this variable is 1 when province p belongs to the treatment group. $Post_{it}$ is a dummy variable indicating whether it is after CMI implementation, and the value of 1 represents that after CMI implementation. $\sigma_{p}$ is a provincial dummy variable, and $Z_{ipt}^{\prime }$ represents other control variables at the individual level to account for confounding indicators at the provincial and individual levels. $\lambda_{t}$ is the time effect and $\varepsilon_{ipt}$ is the error term for the model. In this regression, the coefficient of $treat_{ipt}\times Post_{it}$ is what we focus on as it reflects the effect of CMI.

#### 3.3.2. Fixed-Effects Model

We also adopted a fixed-effects (FE) model to study the impact of CMI to account for the influence of independent variables that did not change over time. The specific model is set as follows:(2)$$y_{ipt}=\alpha +\delta \cdot treat_{ipt}\times Post_{it}+Z_{ipt}^{\prime }\eta +\lambda_{t}+\rho_{i}+\varepsilon_{ipt}$$

In comparison to Equation (1) in the DID model, the variable $treat_{p}$ and the province fixed effect $\sigma_{p}$ are less in the FE model, as in the FE model, these variables are automatically omitted from the estimate. $\rho_{i}$ represents the individual fixed effect, which is used to mitigate the effect of unobservable factors that do not change over time on an individual level. Stata SE 16.0 was used for all analyses (Stata Corp, College Station, TX, USA).

## 4. Results

### 4.1. Characteristics of the Respondents

The characteristics of the elderly in the treatment and control groups are shown in Table 1 for various years. As can be seen, the treatment group had a sample size of 344 (22%) and the control group had a sample size of 1199 (78%), respectively. In 2008, the average age of senior citizens was 75, with 48% of men and 52% of women included in the sample. The majority of the sampled older adults lived in rural areas and participated in the NCMS. Individuals’ health indicators deteriorated as they aged, both subjective and objective health indicators. Between 2008 and 2018, the average number of chronic diseases per person in the treatment group increased from 1 to 1.73, and the proportion of elderly with a serious illness increased from 13% to 38% in the last two years. The average number of chronic diseases in the control group increased from 1 to 1.42, while the proportion of those who had a serious illness in the previous two years increased from 16% to 35%. In terms of healthcare expenditures, both total healthcare expenditures and out-of-pocket expenditures increased year after year in both the treatment and control groups. The treatment group’s total health expenditure increased from 1152.72 CNY in 2008 to 6068.67 CNY in 2018, while the control group’s total health expenditure increased from 1384.51 CNY in 2008 to 6229.12 CNY in 2018. Similar trends can be seen in inpatient and out-of-pocket expenditures.

### 4.2. The Impact of CMI on the Healthcare Expenditures among the Elderly

The parallel trend assumption is the fundamental premise of the DID method. It states that in the absence of policy shocks, the treatment and control groups’ outcome variables should exhibit the same trend of change, thereby excluding any systemic differences between the two groups. Three methods were used to examine the parallel trend hypothesis. To begin, we plotted the trend in healthcare expenditures from 2008 to 2018 for the treatment and control groups (Figure A2). Due to the absence of impatient costs in the CLHLS prior to the 2012 wave, we plotted only changes in total and out-of-pocket healthcare expenditures. As illustrated in the figure, the variation trend in total health expenditures and out-of-pocket health expenditures was essentially identical in the CMI pre-treatment and control groups. With time, a significant difference between the two groups emerged, which partially reflects the implementation effect of CMI. CMI implemented full coverage in 2016, and the gap between the two groups gradually narrowed in 2018, reflecting the impact of increased CMI coverage. Second, we performed a T-test on the difference in healthcare expenditures prior to the policy’s implementation. Prior to CMI, there was no significant difference in healthcare expenditures between the treatment and control groups (*p* > 0.1), as shown in Table A1. Thirdly, we consulted the existing literature [47] and developed a dynamic model to determine whether the parallel trend assumption holds true. The results (Figure A3) indicated that while CMI had no effect on the elderly’s healthcare expenditures prior to the policy’s implementation, all types of healthcare expenditures increased significantly one year later, indicating that the pre-treatment and control groups met the parallel trend hypothesis. Additionally, we plotted the change in the number of medical and technical personnel per thousand population and GDP per capita for treatment and control group provinces from 2008 to 2018 (Figure A4). The results indicated that the two groups of provinces have little in common in terms of economic development, and that the number of medical and technical personnel per thousand population in control provinces prior to the implementation of the CMI was even higher.

In Table 2, panels A and B summarized the regression results for the DID and FE models, respectively (The complete estimation results of DID and FE model are displayed in Appendix A). CMI implementation had the potential to significantly increase the elderly’s healthcare expenditures. After controlling for other covariables affecting healthcare expenditures, the benchmark DID model in Panel A indicated that CMI increased total healthcare expenditures, total OOP expenditures, as well as inpatient and corresponding OOP expenditures (*p* < 0.01). Although the magnitude and significance of the effect were reduced when the individual fixed-effect model was used, the results consistently indicated that CMI significantly increased the elderly’s healthcare expenditures (*p* < 0.05). To be precise, after CMI, the elderly’s logarithmic total health expenditures increased by an average of 0.78 units (*p* < 0.05), while their logarithmic out-of-pocket expenditures increased by 0.65 units (*p* < 0.01). Similarly, CMI increased logarithmic inpatient expenditures of the elderly by 0.58 units (*p* < 0.05) and their out-of-pocket inpatient expenditures by 0.63 units (*p* < 0.05).

### 4.3. The Heterogeneity of Impact of CMI on Healthcare Expenditures

To further examine the heterogeneity of the effect of CMI, samples were stratified by residence, income, and whether the respondent had a serious illness in the preceding two years. The FE model in Equation (2) was conducted, and the results are shown in Table 3 (The complete estimate results are displayed in Appendix A). To begin, the implementation of CMI had a greater impact on rural residents. Following CMI, rural residents’ total logarithmic healthcare expenditures increased by an average of 0.81 units (*p* < 0.01), while their total logarithmic out-of-pocket expenditures increased by 0.64 units (*p* < 0.01). Rural residents’ logarithmic inpatient expenditure increased by 0.61 units, with logarithmic out-of-pocket inpatient expenditure increasing by 0.71 units. However, the implementation of CMI had no significant effect on urban residents’ healthcare expenditures (*p* > 0.1). Second, from an income standpoint, CMI implementation had a greater impact on the high-income group’s healthcare expenditures. CMI had the potential to increase the total logarithmic health expenditure of the high- and low-income groups by 1.05 (*p* < 0.01) and 0.52 units (*p* < 0.1), respectively. After CMI was implemented, the logarithmic out-of-pocket health expenditures of high-income groups increased by 0.91 units on average (*p* < 0.01). Although low-income groups’ logarithmic out-of-pocket expenditures increased by 0.3 units as well, this effect was not statistically significant. CMI increased high- and low-income groups’ logarithmic inpatient expenditures by 0.39 (*p* < 0.1) and 0.58 (*p* < 0.1), respectively. The logarithmic out-of-pocket inpatient expenditures of the high- and low-income groups increased by 0.48 (*p* < 0.05) and 0.51 percentage points, respectively. However, the effect on low-income individuals was not statistically significant (*p* > 0.1). Additionally, both total healthcare expenditures and out-of-pocket spending increased following the implementation of CMI, regardless of whether an individual had a serious illness in the preceding two years, with the coefficient magnitude indicating a greater increase in health expenditures for individuals who had a serious illness in the preceding two years (0.99 vs. 0.73; 1.30 vs. 0.53). CMI increased the logarithmic inpatient expenditures of this group by 1.40 units (*p* < 0.1) but did not significantly increase out-of-pocket inpatient expenditures for the elderly with serious illness in the preceding two years (*p* > 0.1). CMI increased logarithmic inpatient and out-of-pocket expenditures by 0.4 (*p* < 0.1) and 0.48 units (*p* < 0.05), respectively, for patients who had not had a serious illness in the preceding two years.

### 4.4. The Robustness Check of Results

To ensure the reliability and robustness of the regression results, we examined their robustness in three ways. First, some provinces and cities conducted pilot projects in a few counties and cities before gradually expanding them, such as Jiangxi Province. To eliminate the possibility of these provinces interfering with the results, they were excluded and then analyzed using the FE model. Table 4 panel A contains the results (The complete estimate results are displayed in Appendix A). After excluding these provinces (Jiangsu, Anhui, Jiangxi, Guangdong, and Shaanxi) there was no significant difference between them, demonstrating the results’ robustness.

Second, we used baseline characteristics from 2008 to match the treatment and control groups’ propensity scores and retained the matched samples for estimation of the fixed-effects model. Panel B’s results in Table 4 (The complete estimate results are displayed in Appendix A) indicated that there was no significant change in the results following propensity score matching, which is consistent with the results of the fixed-effect estimation in Table 2.

Third, to rule out the possibility that the results were due to chance, we performed the following placebo test. We randomly assigned six provinces to the treatment group and other provinces to the control group in our samples, and then repeated the FE model in Equation (2) 1000 times. At each time point, we obtained the T-statistic of the estimator coefficient. We obtained Figure 2 by plotting the kernel density distribution of these 1000 T-test values. As can be seen, after 1000 simulations, the T-test values for the FE coefficients almost perfectly fit the standard normal distribution, whereas the real T-values were significantly different from the central value. As a result, our results were unaffected by accidental factors, indicating the estimation results’ robustness.

## 5. Discussion

This study examined the effect of CMI on elderly healthcare expenditures using CLHLS data, and the DID and FE models. The findings indicated that CMI increased elderly healthcare expenditures, including total and inpatient expenditures. Simultaneously, CMI increased individual out-of-pocket expenditures. Our findings indicated that the elderly’s total healthcare expenditures and total out-of-pocket healthcare increased significantly following the implementation of CMI, with the increase in total expenditure being greater than the increase in out-of-pocket expenses (0.78 vs. 0.65). Out-of-pocket inpatient expenditures increased more than inpatient expenditures (0.63 vs. 0.58). Health insurance allows insured persons to pay less for medical services and increase their access to healthcare by sharing medical economic burdens, thereby converting medical service needs into effective demand and increasing healthcare expenditures [48]. CMI further decreased the relative cost of medical services and increased healthcare utilization by allowing residents to receive secondary compensation based on URBMI and NCMS. According to Anderson’s health service utilization model, the implementation of CMI expands the enabling resources of health insurance participants and reduces the financial constraints individuals face when seeking medical care. CMI offers alternative opportunities and possibilities for medical treatment for this group of individuals who previously had healthcare needs but could not utilize health services owing to budgetary concerns. This release of previously unmet medical service needs is one of the reasons contributing to the rise in total health expenditures and hospitalization costs.

Individuals’ preference for and pursuit of high-quality health service utilization is another factor contributing to the increase of health expenditures and out-of-pocket expenditures [49]. Implementing CMI can not only alleviate unmet healthcare demands but can also increase individuals’ willingness to utilize higher-quality medical services. This demand for high-quality healthcare would increase individual healthcare expenditures, including out-of-pocket expenditures. Firstly, higher-quality health services are frequently associated with higher price levels. Individuals must spend more than in the past to receive high-quality medical care. Secondly, to better promote the hierarchical diagnostic and treatment system, China implements different reimbursement rates at different levels of health facilities, with the reimbursement rate for high-level health facilities often being lower. In addition, many high-quality health services are not on the list of reimbursable health insurance. Individual healthcare expenditures and out-of-pocket healthcare costs have increased as a result of these causes. Our findings are consistent with a regional CMI assessment study conducted in China [29], which demonstrated that CMI increased hospitalization and outpatient expenses, in part because patients are treated more frequently in high-level health facilities (especially urban tertiary hospitals) than in primary medical facilities (village clinics, township hospitals, or community health service centers) following CMI implementation.

In addition, induced demand from health providers and excessive demand from the demand side may also be potential causes of the increase in health expenditures. From a supply-side perspective, doctors prescribe more expensive prescriptions in high-level health facilities because they have access to higher-quality treatment, resulting in increased healthcare expenditures for individuals receiving treatment [29]. The impact of pre-implementation of low-income and credit constraints on patient utilization of medical services has been mitigated to some extent following the implementation of CMI. They are more likely and willing to undergo additional examinations and treatment recommended by doctors, and they are more willing to follow doctors’ advice to stay in the hospital and generate additional healthcare expenditures than previously [29]. Additionally, studies have shown that physicians are more motivated to prescribe more drugs and treatments under a fee-for-service payment system [50,51,52]. The moral hazard model predicts that health insurance participants will incur lower medical costs as a result of the co-payment mechanism when utilizing medical services, which will result in an increase in health service needs [53,54]. Health insurance coverage and reimbursement ratios have been shown to result in insured individuals using more health services than is necessary [39,55,56,57,58]. By increasing reimbursement from CMI, the insured’s proportion of co-payments is reduced further, thereby improving healthcare utilization and the acceptable price of medical services.

Heterogeneity analysis revealed that CMI had a greater effect on rural residents in terms of total medical expenditures, total OOP expenditures, inpatient expenditures, and inpatient OOP expenditures. However, for those who live in cities or towns, the implementation of CMI had no effect on their healthcare expenditures. This effect could be explained by the fact that URBMI and NCMS have different medical insurance designs. Rural residents voluntarily participate in NCMS, while urban unemployed residents participate in URBMI, according to the insured policy. NCMS focuses on compensating insured persons for hospitalization expenses, whereas URBMI focuses on compensating insured persons for hospitalization and catastrophic outpatient care expenses. Since URBMI compensates insured persons for a greater proportion of inpatient medical expenses than NCMS [5], the elderly living in rural areas face higher out-of-pocket medical expenses. According to one study, even after receiving compensation from the NCMS, the majority of families with catastrophic health expenditure continue to face catastrophic health expenditure pressure [59]. As a result of the CMI’s implementation, the reimbursement rate for rural residents’ medical expenses has been increased further, and the healthcare utilization restricted by the NCMS’s low reimbursement rate has been improved. Simultaneously, rural residents would face higher out-of-pocket costs as a result of higher-quality and more expensive hospital care [27]. CMI also increased total healthcare expenditures and out-of-pocket expenses for elderly living in cities or towns, but the effects were not statistically significant. CMI is unlikely to have an effect on this group’s health expenditures due to URBMI’s more compensation lists and higher reimbursement rates than NCMS.

Income heterogeneity analysis revealed that CMI increased total healthcare expenditures for both high- and low-income residents, but more for high-income residents. However, in terms of hospitalization, CMI has a greater positive effect on low-income groups. CMI increased the high-income group’s total OOP and inpatient OOP expenditures, but did not significantly increase low-income groups’ OOP expenditures, indicating CMI’s beneficial effect on inpatient utilization and financial protection for low-income groups. According to one study, patients with the lowest income have the greatest reduction in OOP following CMI, and this reduction will continue to decrease over the implementation period of the CMI [23]. CMI is more protective for low-income individuals, which may also be explained by their preference for health facilities when ill. Wagstaff et al. discovered that the poor are more likely to seek treatment at low-level health facilities, putting less strain on their OOP expenditures [60]. Although, some research indicated that high OOP costs limit the ability of people in impoverished economic situations to benefit from the CMI and erode the CMI’s fairness [28]. CMI considers the needs of low-income residents during the gradual implementation process. Not only are poor households with filing cards exempt from health insurance premiums, but they also have a lower deductible than ordinary residents, and the proportion of segmented reimbursement is also higher than the general population [61], ensuring the CMI’s fairness.

Individuals suffering from serious diseases frequently face financial hardship due to high out-of-pocket expenses and the loss of human health capital. As a result, they are at risk of becoming trapped in a vicious cycle of disease and poverty [62]. Our study discovered that CMI had an effect on the healthcare expenditures of the elderly regardless of whether they had been diagnosed with a serious disease in the preceding two years, and this effect is more pronounced in people with serious illness. Due to the characteristics of the CMI segmented reimbursement ratio, individuals with higher healthcare expenditures would receive a higher level of compensation. For elderly with a serious illness in the preceding two years, the positive effect of CMI on inpatient expenditures reflected the promotion effect of CMI on healthcare utilization, whereas the insignificant effect of CMI on inpatient OOP expenditures reflected the protective effect of CMI on their financial burden, which is consistent with previous research findings [22,23]. However, CMI did not significantly improve the financial burden of patients with serious illness in terms of total healthcare expenditure, which may be due to reimbursement limitations [24]. Although CMI provided additional compensation, the compensation list is consistent with that of basic medical insurance, and for patients with serious illnesses, the drugs and treatments used to treat them may be excluded from the compensation list, necessitating higher out-of-pocket healthcare expenditures [24].

Our research has policy implications. To begin, in terms of compensation design, the CMI compensation list should be appropriately broadened, not limited to coverage consistent with basic medical insurance, and dynamic adjustments should be made to account for regional characteristics and the specific use of medical drugs and treatments. Second, it is necessary to pay attention to potentially poor individuals who relapse into poverty owing to disease, such as low-income residents and patients who previously suffered a serious illness. By establishing a more precise compensation scheme, it is possible to accurately ensure reimbursement for the insured who needs CMI. It is essential and important to improve the precision of policy implementation. Third, in addition to focusing on hospitalization expenditures, specific diseases associated with high healthcare costs can be included in the scope of CMI reimbursement to bolster CMI’s financial protective effect.

Our research was limited in several ways. First, because there are no data on healthcare utilization in CLHLS, we could not determine the effect of CMI on elderly healthcare utilization (such as the number of hospitalizations). Second, because the CLHLS survey was self-reported, self-reporting errors were unavoidable. Third, we could not estimate the behavior changes of suppliers and demanders following the implementation of CMI due to the limited data, which may not be observed for excessive demand and underutilized health services. Therefore, despite that we provided the hypothesis that the behavior of the supply side and the demand side may result in an increase in health expenditures, we are unable to prove it. Verification of this assumption will be investigated in the subsequent stage. At the same time, we could not adjust the survival differences between elderly individuals in the sample for healthcare utilization and accessibility, which resulted in some potential bias in our estimation results. It should be noted that CMI is a health insurance scheme whose primary purpose is to lower the catastrophic health expenditures of urban and rural Chinese residents. The impact of CMI on the incidence of catastrophic health expenditure (CHE) must be studied. We could not quantify the prevalence of CHE in households using CLHLS data, however, due to the limited availability of data. The impact of CMI impact coverage on the incidence of CHE in households will be our next study topic.

## 6. Conclusions

CMI increased total and out-of-pocket healthcare expenditures significantly, as well as inpatient and corresponding out-of-pocket expenditures. CMI had a greater impact on rural elderly. Economic burden protection has been enhanced for low-income groups and patients with serious diseases over the last two years. To amplify the protective effect of CMI, targeted policies and measures such as expanding the CMI compensation list, establishing a more precise compensation scheme, and specific diseases associated with high healthcare expenditures can be included in the scope of CMI reimbursement.

## Figures and Tables

**Figure 1 ijerph-19-14313-f001:**
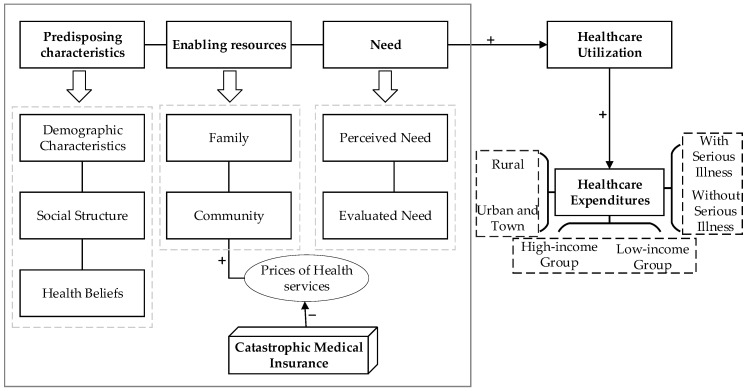
Conceptual framework.

**Figure 2 ijerph-19-14313-f002:**
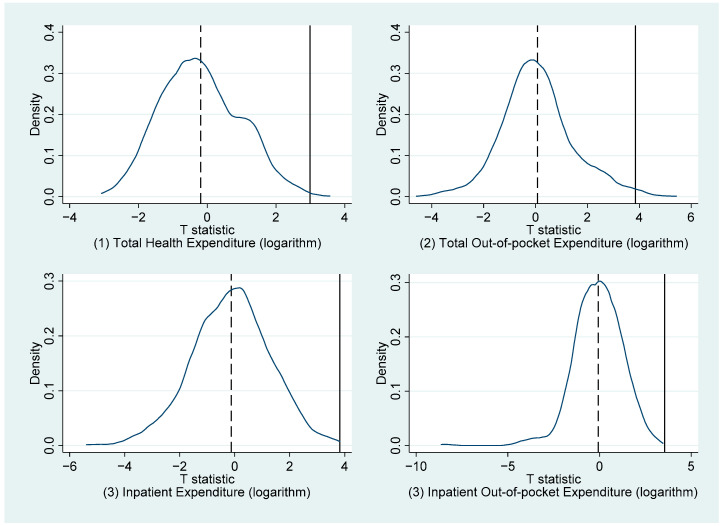
Placebo test.

**Table 1 ijerph-19-14313-t001:** Summary statistics.

	Year = 2008		Year = 2012		Year = 2014		Year = 2018	
Variable	Control	Treatment	Control	Treatment	Control	Treatment	Control	Treatment
Total Medical Expenditure	1384.51 (5254.24)	1152.72 (2370.71)	2890.66 (8163.63)	2983.13 (8203.16)	4121.17 (9963.70)	5920.37 (13,754.66)	6229.12 (15,575.89)	6068.67 (13,916.62)
Total Out-of-pocket Expenditure	906.78 (2501.99)	835.89 (1380.81)	1826.86 (4920.07)	1779.33 (3854.65)	1991.54 (5354.93)	3157.59 (9891.93)	3563.01 (10,017.50)	4008.83 (11,087.67)
Inpatient Expenditure	-	-	1518.92 (5849.45)	1446.51 (6812.03)	2508.52 (8154.63)	2900.57 (8992.01)	4001.67 (11,854.85)	3580.30 (10,892.72)
Inpatient Out-of-pocket Expenditure	-	-	902.32 (3873.50)	560.41 (1984.51)	1075.40 (4436.27)	1539.98 (7136.82)	2340.41 (8812.12)	2232.13 (8377.17)
Age	74.99 (8.12)	74.91 (7.97)	78.03 (8.15)	77.90 (7.98)	80.72 (8.13)	80.70 (7.99)	84.84 (8.09)	84.84 (7.90)
Gender								
Male	580 (48%)	145 (42%)	580 (48%)	145 (42%)	580 (48%)	145 (42%)	580 (48%)	145 (42%)
Female	619 (52%)	199 (58%)	619 (52%)	199 (58%)	619 (52%)	199 (58%)	619 (52%)	199 (58%)
Years of education	2.98 (3.68)	2.52 (3.59)	2.98 (3.68)	2.52 (3.59)	2.98 (3.68)	2.52 (3.59)	2.98 (3.68)	2.52 (3.59)
Residence								
Rural	1054 (88%)	290 (84%)	1019 (85%)	287 (83%)	1021 (85%)	273 (79%)	995 (83%)	265 (77%)
Urban or town	145 (12%)	54 (16%)	180 (15%)	57 (17%)	178 (15%)	71 (21%)	204 (17%)	79 (23%)
Married								
No	443 (37%)	136 (40%)	513 (43%)	151 (44%)	570 (48%)	169 (49%)	672 (56%)	204 (59%)
Yes	756 (63%)	208 (60%)	686 (57%)	193 (56%)	629 (52%)	175 (51%)	527 (44%)	140 (41%)
Number of Children alive	3.91 (1.68)	3.59 (1.67)	3.31 (2.57)	3.37 (2.26)	3.73 (1.71)	3.60 (1.61)	3.70 (1.74)	3.42 (1.66)
Live alone								
No	1029 (86%)	280 (81%)	1003 (84%)	272 (79%)	979 (82%)	262 (76%)	1010 (84%)	272 (79%)
Yes	170 (14%)	64 (19%)	196 (16%)	72 (21%)	220 (18%)	82 (24%)	189 (16%)	72 (21%)
Health Insurance								
NCMS	1037 (86%)	276 (80%)	1013 (84%)	275 (80%)	964 (80%)	259 (75%)	925 (77%)	279 (81%)
URBMI	162 (14%)	68 (20%)	186 (16%)	69 (20%)	235 (20%)	85 (25%)	274 (23%)	65 (19%)
Logarithm of household income per capita	8.15 (1.23)	8.23 (0.95)	8.38 (1.48)	8.69 (1.36)	8.66 (1.29)	8.95 (1.46)	8.73 (1.66)	9.30 (1.29)
Self-rated health								
Very Good	180 (15%)	38 (11%)	146 (12%)	28 (8%)	124 (10%)	31 (9%)	126 (11%)	37 (11%)
Good	467 (39%)	160 (47%)	425 (35%)	118 (34%)	390 (33%)	133 (39%)	376 (31%)	107 (31%)
Fair	366 (31%)	108 (31%)	440 (37%)	154 (45%)	480 (40%)	126 (37%)	458 (38%)	128 (37%)
Poor	158 (13%)	33 (10%)	168 (14%)	40 (12%)	171 (14%)	50 (15%)	140 (12%)	44 (13%)
Very Poor	28 (2%)	5 (1%)	20 (2%)	4 (1%)	34 (3%)	4 (1%)	99 (8%)	28 (8%)
Number of chronic diseases	1.00 (1.13)	0.99 (1.32)	1.23 (1.38)	1.49 (1.52)	1.26 (1.35)	1.68 (1.52)	1.42 (1.46)	1.73 (1.67)
Serious Illness in the past 2 years								
No	1010 (84%)	299 (87%)	914 (76%)	273 (79%)	866 (72%)	255 (74%)	783 (65%)	213 (62%)
Yes	189 (16%)	45 (13%)	285 (24%)	71 (21%)	333 (28%)	89 (26%)	416 (35%)	131 (38%)

Note: the health expenditures were standardized to 2008 Chinese Yuan (CNY) using the healthcare price index from the Chinese Statistical Yearbook 2009–2019, and the household income per capita was standardized to 2008 Chinese Yuan (CNY) using the consumer price index from the Chinese Statistical Yearbook 2009–2019.

**Table 2 ijerph-19-14313-t002:** The impact of CMI on healthcare expenditure.

	(1)	(2)	(3)	(4)
Variables	Logarithm Total Medical Expenditure	Logarithm Total OOP Expenditure	Logarithm Inpatient Expenditure	Logarithm Inpatient OOP Expenditure
Panel A: DID analysis				
$treated_{ipt}\times Post_{it}$	0.779 ***	0.681 ***	0.669 ***	0.751 ***
	(0.260)	(0.177)	(0.175)	(0.211)
Covariates	Yes	Yes	Yes	Yes
N	4629	4629	3086	3086
${\bar{R}}^{2}$	0.255	0.214	0.416	0.380
Panel B: FE analysis				
$treated_{ipt}\times Post_{it}$	0.776 **	0.649 ***	0.577 **	0.625 **
	(0.310)	(0.172)	(0.225)	(0.237)
Covariates	Yes	Yes	Yes	Yes
N	4629	4629	3086	3086
${\bar{R}}^{2}$	0.155	0.132	0.319	0.303

Note: *** *p* < 0.01, ** *p* < 0.05. Robust standard errors clustered at the province level are reported in the parentheses. In addition to the covariates listed in Section 3.2.3, the model also controls the patient’s diagnostic information, namely a series of dummy variables including hypertension, diabetes, heart disease, stroke and cerebrovascular diseases, bronchitis, emphysema, asthma or pneumonia, tuberculosis, cataracts, glaucoma, cancer, prostate disease, gastrointestinal ulcers, Parkinson’s disease, bedsores, arthritis, dementia, epilepsy, cholecystitis or cholelithiasis, dyslipidemia, rheumatic or rheumatoid arthritis, chronic nephritis, hyperplasia of mammary glands, uterine fibroids, benign prostatic hyperplasia, and hepatitis.

**Table 3 ijerph-19-14313-t003:** The impact of CMI on healthcare expenditures among different subgroups.

	(1)	(2)	(3)	(4)
Subgroups	Logarithm Total Medical Expenditure	Logarithm Total OOP Expenditure	Logarithm Inpatient Expenditure	Logarithm Inpatient OOP Expenditure
Panel A: Residence				
Rural:	0.810 ***	0.641 ***	0.607 **	0.704 ***
	(0.214)	(0.220)	(0.274)	(0.252)
Urban or Town:	0.698	0.380	−0.232	−0.440
	(0.518)	(0.536)	(0.593)	(0.514)
Panel B: Income				
Low-income group:	0.515 *	0.301	0.575 *	0.507
	(0.275)	(0.289)	(0.316)	(0.334)
High-income group:	1.046 ***	0.912 ***	0.385 *	0.483 **
	(0.269)	(0.272)	(0.230)	(0.210)
Panel C: Serious Illness				
Without Serious Illness:	0.726 ***	0.534 **	0.403 *	0.476 **
	(0.246)	(0.255)	(0.208)	(0.210)
With Serious Illness:	0.989 **	1.297 ***	1.398 *	1.216
	(0.448)	(0.494)	(0.737)	(0.953)

Note: *** *p* < 0.01, ** *p* < 0.05, * *p* < 0.1. Robust standard errors clustered at the province level are reported in the parentheses. In addition to the covariates listed in Section 3.2.3, the model also controls the patient’s diagnostic information, namely a series of dummy variables including hypertension, diabetes, heart disease, stroke and cerebrovascular diseases, bronchitis, emphysema, asthma or pneumonia, tuberculosis, cataracts, glaucoma, cancer, prostate disease, gastrointestinal ulcers, Parkinson’s disease, bedsores, arthritis, dementia, epilepsy, cholecystitis or cholelithiasis, dyslipidemia, rheumatic or rheumatoid arthritis, chronic nephritis, hyperplasia of mammary glands, uterine fibroids, benign prostatic hyperplasia, and hepatitis.

**Table 4 ijerph-19-14313-t004:** Robustness check.

	(1)	(2)	(3)	(4)
Variables	Logarithm Total Medical Expenditure	Logarithm Total OOP Expenditure	Logarithm Inpatient Expenditure	Logarithm Inpatient OOP Expenditure
Panel A: Excluding the confounding provinces				
$treated_{ipt}\times Post_{it}$	0.752 **	0.588 ***	0.588 **	0.652 **
	(0.328)	(0.197)	(0.257)	(0.251)
Covariates	Yes	Yes	Yes	Yes
N	3711	3711	2474	2474
${\bar{R}}^{2}$	0.164	0.133	0.308	0.293
Panel B: Matching-based fixed-effects model				
$treated_{ipt}\times Post_{it}$	0.773 **	0.645 ***	0.581 **	0.621 **
	(0.317)	(0.180)	(0.229)	(0.241)
Covariates	Yes	Yes	Yes	Yes
N	4605	4605	3070	3070
${\bar{R}}^{2}$	0.156	0.133	0.322	0.302

Note: *** *p* < 0.01, ** *p* < 0.05. Robust standard errors clustered at the province level are reported in the parentheses. In addition to the covariates listed in Section 3.2.3, the model also controls the patient’s diagnostic information, namely a series of dummy variables including hypertension, diabetes, heart disease, stroke and cerebrovascular diseases, bronchitis, emphysema, asthma or pneumonia, tuberculosis, cataracts, glaucoma, cancer, prostate disease, gastrointestinal ulcers, Parkinson’s disease, bedsores, arthritis, dementia, epilepsy, cholecystitis or cholelithiasis, dyslipidemia, rheumatic or rheumatoid arthritis, chronic nephritis, hyperplasia of mammary glands, uterine fibroids, benign prostatic hyperplasia, and hepatitis.

## Data Availability

The data that support the findings of this study are available in the Chinese Longitudinal Healthy Longevity Survey at https://sites.duke.edu/centerforaging/programs/chinese-longitudinal-healthy-longevity-survey-clhls/ (accessed on 30 October 2022) [43].

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
