# Peer review of "Healthcare Expenditures among the Elderly in China: The Role of Catastrophic Medical Insurance"

_ijerph, 2022, doi:10.3390/ijerph192114313_

Round 1
Reviewer 1 Report
1. The contribution and innovation points are not outstanding enough. What is the significance of the heterogeneity analysis?
2. Figure 1 "Conceptual framework" is not presented clearly, in which heterogeneities are not reflected. "Health service demand" (effective or excessive)is not contained in your data analysis.
3. Since the policy shocks of CMI are all in 2013 rather than at multiple points in time, it is better to write treatedipt as an interaction term in model (1).
4. The contents of line 197-210 (3.2.3.Covariates) and line 225-238 (the beginning of 3.3.2. Fixed effects model) are completely identical.
Author Response
Please see the attachmemt.

Reviewer 2 Report
Please see the attachment.

Reviewer 3 Report
This is a relatively mature empirical study. Both the format and content of the paper meet the requirements of a scientific paper, the method selected is appropriate, and the conclusion is clear. The paper meets the requirements for publication and is recommended for publication. But the authors also need to pay attention to several issues.
1. The research hypothesis is usually not expressed as the effect of XX on XX is unknown, therefore, hypothesis 1 needs to be revised.
2. Some abbreviations in the text are not indicated when they are used for the first time, such as OOP.
3. The format of some quotations is not correct, such as line86-87.
4. Although the authors have put forward and tested hypotheses about Catastrophic Medical Insurance for different types of elders and elders in different regions, what are the theoretical implications behind these conclusions? The authors do not seem to think further, as scholars from China's top-ranked universities should not only draw empirical research conclusions, but also consider the theoretical issues behind the empirical conclusions. Because of this lack of attention, the contribution of this study is greatly reduced.
5.The tables in the text only reported the results for the main variables, the results for the full model should be attached as an supplementary files.
Round 2
Reviewer 1 Report
It is pleasing to see that the authors has made more adequate improvements and clarifications based on the reviewers' recommendations. Overall, this version presents a relatively normative quantitative study. To further impove your manuscript, I would like to propose a few small points for reference.
1. Parts of Conceptual framework (Figure 1) are still ambiguous. For example, "Predisposing characteristics" should contian three secondary aspects ("Demographic Characteristics", "Social Structure", and "Health Beliefs"). In your figure, the subordinate relationship is not clear enough.
2. Pay more attention to the consistency of tenses. (e.g. the part of hypothesis)
Reviewer 2 Report
This manuscript has been significantly improved
Author Response
Thank you for your review.